# The Different Metabolic Responses of Resistant and Susceptible Wheats to *Fusarium graminearum* Inoculation

**DOI:** 10.3390/metabo12080727

**Published:** 2022-08-06

**Authors:** Caixiang Liu, Fangfang Chen, Laixing Liu, Xinyu Fan, Huili Liu, Danyun Zeng, Xu Zhang

**Affiliations:** 1State Key Laboratory of Magnetic Resonance and Atomic and Molecular Physics, National Center for Magnetic Resonance in Wuhan, Wuhan Institute of Physics and Mathematics, Innovation Academy for Precision Measurement of Science and Technology, Chinese Academy of Sciences, Wuhan 430071, China; 2University of Chinese Academy of Sciences, Beijing 100049, China; 3Songjiang Yujian High School affiliated to Shanghai Foreign Language School, Shanghai 201600, China; 4Molecular Biotechnology Laboratory of Triticeae Crops, Huazhong Agricultural University, Wuhan 430070, China; 5School of Management Wuhan Institute of Technology, Wuhan 430205, China; 6State Key Laboratory of Component-based Chinese Medicine, Tianjin University of Traditional Chinese Medicine, Tianjin 300193, China; 7Wuhan National Laboratory for Optoelectronics, Huazhong University of Science and Technology, Wuhan 430074, China

**Keywords:** *Fusarium graminearum*, resistant and susceptible wheat, trehalose biosynthesis, *TPS1*
^−^, *TPS2*
^−^, metabonomics, NMR

## Abstract

Fusarium head blight (FHB) is a serious wheat disease caused by Fusarium graminearum (*Fg*) Schwabe. FHB can cause huge loss in wheat yield. In addition, trichothecene mycotoxins produced by *Fg* are harmful to the environment and humans. In our previous study, we obtained two mutants *TPS1*^−^ and *TPS2*^−^. Neither of these mutants could synthesize trehalose, and they produced fewer mycotoxins. To understand the complex interaction between *Fg* and wheat, we systematically analyzed the metabolic responses of FHB-susceptible and -resistant wheat to ddH_2_O, the *TPS*^−^ mutants and wild type (WT) using NMR combined with multivariate analysis. More than 40 metabolites were identified in wheat extracts including sugars, amino acids, organic acids, choline metabolites and other metabolites. When infected by *Fg*, FHB-resistant and -susceptible wheat plants showed different metabolic responses. For FHB-resistant wheat, there were clear metabolic differences between inoculation with mutants (*TPS1*^−^/*TPS2*^−^) and with ddH_2_O/WT. For the susceptible wheat, there were obvious metabolic differences between inoculation with mutant (*TPS1*^−^/*TPS2*^−^) and inoculation with ddH_2_O; however, there were no significant metabolic differences between inoculation with *TPS*^−^ mutants and with WT. Specifically, compared with ddH_2_O, resistant wheat increased the levels of Phe, p-hydroxy cinnamic acid (p-HCA), and chlorogenic acid in response to *TPS*^−^ mutants; however, susceptible wheat did not. Shikimate-mediated secondary metabolism was activated in the FHB-resistant wheat to inhibit the growth of *Fg* and reduce the production of mycotoxins. These results can be helpful for the development of FHB-resistant wheat varieties, although the molecular relationship between the trehalose biosynthetic pathway in *Fg* and shikimate-mediated secondary metabolism in wheat remains to be further studied.

## 1. Introduction

Fusarium head blight (FHB) is one of the most devastating diseases of wheat (*Triticum aestivum* L.) globally. FHB, caused by *Fusarium graminearum* (*Fg*) Schwabe (teleomorph Gibberella zeae Petch), not only leads to huge reductions in wheat grain yield, but also harms the environment and humans by producing deteriorated grain quality contaminated with trichothecene mycotoxins [1,2,3,4]. Since 1993, FHB has become a major problem for the agriculture industry in North America [5,6]. Apart from FHB, *Fg* can also infect other cereals (such as barley, maize, and oats), and cause stalk rot or root rot [7]. Although fungicides have been applied to control *Fg*, the resulting environmental problems and fungicide resistance are not negligible [7]. Breeding FHB-resistant wheat varieties is considered to be an economical and environmentally friendly approach to managing FHB. 

Trehalose is a nonreducing disaccharide formed by two glucose molecules linked with a 1α–1α bond, and is widely found in plants, bacteria, fungi and insects [8]. In recent years, trehalose has drawn considerable attention for its important functions in serving as a carbon source [9], regulating osmotic pressure as a compatible solute in prokaryotes [10], and stabilizing and protecting membranes and proteins [11,12]. In addition, trehalose plays a significant role in the response to various stresses such as oxidative stress, heat, and drought [13,14,15]. More importantly, trehalose may play a role in signaling or regulation [16]. 

It is reported that there are at least five pathways for trehalose biosynthesis in different organisms [8,16]. The best-known pathway involves two steps: the first step is being catalyzed by trehalose 6-phosphate synthase (TPS1), and the second step is being catalyzed by trehalose 6-phosphate phosphatase (TPS2). TPS1 catalyzes the transfer of combined uridinediphospho (UDP) glucose and glucose 6-phosphate to generate trehalose 6-phosphate (T6P), while TPS2 is responsible for catalyzing the dephosphorylation of T6P to form trehalose [17,18]. Many studies have demonstrated that, apart from involvement in trehalose synthesis, *TPS* genes also take part in the development, pathogenicity and stress responses in yeast and higher fungi [19]. Blocking trehalose synthesis may be a promising approach for managing fungal diseases [20].

We obtained two mutant strains, *TPS*1^−^ and *TPS2*^−^, from our previous study, carrying a single deletion of *TPS1* or *TPS2*, respectively [21]. The results showed that *TPS1* appeared unessential for *Fg* development and virulence, while *TPS2* deletion abolished sporulation and sexual reproduction of *Fg*. In addition, it was reported that the *TPS2*^−^ mutant had a more significant reduction in the production of mycotoxins compared with the *TPS*1^−^ mutant [22]. 

Metabonomics has emerged as a powerful tool for studying the metabolic responses of plants to both biotic and abiotic stresses [23,24,25,26,27,28,29]. It has been used in understanding the interaction between plants and pathogens [30,31] and between plants and insects [24,32,33]. For example, metabolomics analysis of wheat leaves and stem tissues indicated that the levels of betaine, sucrose, glucose, glutamate, glutamine, alanine, trans-aconitic acid, and some aromatic compounds were positively correlated with FHB resistance [34]. Moreover, Liu et al. [25] found that the activation of γ- amino butyric acid shunt and shikimate-mediated secondary metabolism was vital for rice plants to resist insect infestation. Furthermore, combined transcriptomic and metabolomic analyses revealed that the tryptophan synthesis pathway plays an important role in the resistance of cotton to *V. dahlia* [35]. So far, the metabolic responses of FHB-resistant and -susceptible wheat to *TPS*^−^ mutants and WT is unclear. However, this information can afford us metabolites or metabolic pathways related to FHB resistance and can afford help in controlling *Fg* and developing FHB-resistant wheat varieties.

In this study, we analyzed the metabolomics profiles of FHB-resistant and -susceptible wheat varieties inoculated with ddH_2_O, WT, and *TPS*^−^ mutants. Our objectives are to obtain the different metabolic responses of resistant and susceptible wheat to ddH_2_O, WT, and *TPS*^−^ mutants, which will offer important information for further cultivating FHB-resistant wheat varieties.

## 2. Materials and Methods

### 2.1. Chemicals

Methanol, NaH_2_PO_4_.2H_2_O, and K_2_HPO_4_.3H_2_O were purchased from Guoyao Chemical Co. Ltd. (Shanghai, China), while sodium 3-trimethlysilyl [2, 2, 3, 3-D_4_] propionate (TSP) and D_2_O (99.9% D) were obtained from Cambridge Isotope Laboratory (Miami, FL, USA). 

### 2.2. Fungus Material Culture and Plant Materials

*Fg* strain 5035 (wild type, WT) was isolated from a scabby wheat spike in Wuhan (China). Strain 5035 was highly pathogenic to wheat through producing many mycotoxins [36,37]. *TPS1*^−^ and *TPS2*^−^ are two isogenic strains obtained from homologous recombination *of Fg* strain 5035 through *Agrobacterium*-mediated transformation [21]. Molecular characterization confirmed that trehalose 6-phosphate synthase gene was deleted in the *TPS1*^−^ mutant, while trehalose 6-phosphate phosphatase was deleted in the *TPS2*^−^ mutant [21]. *Fg* strains were cultured in CMC broth (7.5 g/L of carboxymethyl cellulose, 0.5 g/L of KH_2_PO_4_, 0.5 g/L of NH_4_NO_3_, 0.25 g/L of MgSO_4_.7H_2_O, and 0.5 g/L of yeast extract) [38] at 28 °C for 5 days (200 rpm). Conidiaspores were collected and adjusted to the concentration of about 1 × 10^6^ spores/mL, and 10 μL of the conidia was cultured at 28 °C on potato-dextrose agar (PDA) for 3 days. 

In this experiment, the wheat variety Sumai 3, which is resistant to FHB, and Annong 8455, which is susceptible to FHB, were grown in fields at Huazhong Agricultural University, Wuhan, China [39]. At the early anthesis, one spike per wheat was inoculated with 10 μL of ddH_2_O or the above conidia suspensions (*Fg* WT, *TPS1*^−^, and *TPS2*^−^) using a pipette tip for 96 h. Five inoculated spikes were collected as a sample, and a total of 46 samples of the two wheat varieties were obtained to afford five to seven biological replicates (n = 5–7). The wheat spikes were harvested and snap-frozen in liquid nitrogen, then stored at −80 °C until further analysis. 

### 2.3. Metabolite Extraction for Wheat

Metabolites in wheat spikes were extracted using a previously reported method with some improvements [24]. The samples were freeze-dried, ground into powder with a mortar and pestle, and about 25 mg of the powder samples was transferred into a 2 mL Eppendorf tube with the addition of 1.2 mL of methanol/water solution (*v*/*v* = 2/1, −40 °C) and one 5 mm tungsten carbide bead (Qiagen, Germany). The mixture was homogenized using a tissuelyser (Qiagen, Germany) after drastically vortexing for 30 s followed by 15 min intermittent sonication (i.e., 30 s sonication with 30 s break) in an ice bath. The supernatant of each sample was transferred into a new 5 mL Eppendorf tube following centrifugation for 10 min (16,099× *g*, 4 °C). The remaining residues were further extracted twice using the same method, and three supernatants were combined as one sample. After removal of methanol under vacuum, samples were lyophilized. The freeze-dried extracts were redissolved in 600 μL of phosphate buffer (0.1 M K_2_HPO_4_-NaH_2_PO_4_, pH 7.4) containing 50% D_2_O (*v*/*v*) and 0.02% TSP [40]. After being centrifuged for 10 min (16,099× *g*, 4°C), a total of 500 μL of supernatant for each sample was transferred into 5 mm NMR tubes for NMR-based metabolite analysis.

### 2.4. NMR Measurements

All ^1^H NMR spectra of samples for wheat spikes were acquired at 298 K using an inverse detection cryogenic probe on a Bruker AVIII 600 spectrometer (Bruker Biospin, GmbH, Rheinstetten, Germany). The ^1^H NMR spectra were acquired using NOESY pulse sequence (RD-90°-t1-90°-tm-90°-acquisition) with 90° pulse length of about 9.4 μs and t1 was set to 2 μs. Water peak was saturated with a weak irradiation during the recycle delay (RD) of 2 s and a mixing time (tm) of 100 ms, and 32 transients were collected into 64 k data points with a spectral width of 20 ppm. A series of ^2^D NMR spectra including ^1^H-^1^H TOCSY, ^1^H-^1^H COSY, ^1^H-JRES, ^1^H-^13^C HSQC, and ^1^H-^13^C HMBC spectra were acquired using selected samples for metabolite assignments [24].

### 2.5. Spectral Processing and Multivariate Data Analysis

Following phase and baseline correction using TopSpin (v3.1, Bruker Biospin GmbH, Germany), all NMR spectra were referenced to the internal standard TSP at *δ* 0.000 ppm. The spectral region between 0.5 and 10.0 ppm was divided into bins with a width of 0.004 ppm (2.4 Hz) using the AMIX software (v 3.8.3, Bruker Biospin GmbH, Germany). The water regions at δ 4.700–5.100 were removed. A total of 2275 bins for all remaining regions were normalized to the dry weight of wheat spikes to give a dataset in the form of signal area (metabolite quantity) per gram dry weight of wheat.

Principal component analysis (PCA) and orthogonal projection to latent structures discriminant analysis (OPLS-DA) [41] were both performed on the normalized NMR data using SIMCA-P+ software (v12.0, Umetrics, Umea, Sweden). In OPLS-DA models, one orthogonal and one predictive component were calculated using the unit-variance (UV) scaled NMR data as X-matrix and the class information as Y-matrix. The model qualities were described by the explained variances for X-matrix (R^2^X values) and the model predictability (Q^2^ values) with further assessment with ANOVA of the cross-validated residuals (CV-ANOVA) approach where intergroup differences were considered as significant with *p* value < 0.05 [42]. Leave-One-Out (LOO) validation was used in the cross-validation of the models. The results were exhibited in both the form of scores plots and loadings plots, in which scores plots and loadings plots showed group clustering and indicated variables (metabolite levels) contributing to inter-group differences, respectively. In such loading plots, variables were color-coded according to absolute values of the correlation coefficients (|r|) [43], and variables (i.e., metabolite contents) with a cool color (e.g., blue) showed less significant contributions to inter-group differences than those with a warm color (e.g., red). In this study, the metabolites showing statistically significant changes were obtained at the level of *p* < 0.05.

## 3. Results

### 3.1. Metabolic Profiles for FHB-Susceptible and -Resistant Wheats 

^1^H NMR spectra of wheat spike extracts showed obvious difference in the metabolic profiles for FHB-resistant wheat (Sumai 3) inoculated with ddH_2_O, WT, *TPS1*^−^, and *TPS2*^−^ (Figure 1). Similarly, there was also an obvious difference in the metabolic profiles for the FHB-susceptible wheat (Annong 8455) inoculated with ddH_2_O, WT, *TPS1*^−^, and *TPS2*^−^ (Figure 2). Signals were assigned to individual metabolites (Appendix A) based on data in the literature [23,44,45] and in-house databases. A series of 2D NMR spectra were acquired for selected samples to further confirm metabolite identifications. More than 40 metabolites were identified, including 5 sugars (sucrose, glucose, raffinose, fructose, and myo-inositol), 16 amino acids and their metabolites (Val, Ieu, Ile, Thr, Ala, Arg, Met, GABA, Glu, Gln, Asp, Asn, Phe, Trp, Tyr, and His), 9 organic acids (acetate, lactate, pyruvate, succinate, fumarate, citrate, α-ketoglutarate, malate, and formate), 5 choline metabolites (choline, phosphocholine, glycine betaine, ethanolamine, and dimethylglycine), 6 nucleotide metabolites (adenosine, uridine, guanosine, hypoxanthine, inosine, and AMP), and 2 secondary metabolites (p-hydroxy cinnamic acid and chlorogenic acid) (Figure 1 and Figure 2, Appendix A). 

Visual inspection of Figure 1 suggested that for the resistant wheat Sumai 3, inoculation with the *TPS1*^−^ mutant for 96 h induced significant elevation in guanosine level along with a decrease in α-ketoglutarate level when compared with inoculation with ddH_2_O, and it had decreases in myo-inositol and Asp levels when compared with inoculation with WT (Figure 1a–c). In addition, for the resistant wheat Sumai 3, inoculation with the *TPS2*^−^ mutant induced significantly higher levels of Phe and chlorogenic acid than inoculation with ddH_2_O, and it had a lower level of sucrose than with inoculation with WT (Figure 1a,b,d). 

For the susceptible wheat variety Annong 8455, the level of adenosine was higher in response to inoculation with the *TPS1*^−^ mutant for 96 h than inoculation with ddH_2_O, and the levels of sucrose and guanosine were lower after being inoculated with the *TPS2*^−^ mutant compared with inoculation with ddH_2_O (Figure 2). Moreover, inoculation with the *TPS2*^−^ mutant induced a more significant reduction in glucose and malate levels in the susceptible wheat than inoculation with the *TPS1*^−^ mutant (Figure 2c,d). To obtain more detailed information about metabolic changes in the resistant and susceptible wheats induced by ddH_2_O, WT, *TPS1*^−^, and *TPS2*^−^, multivariate data analyses were performed on the NMR data of these wheat spikes.

### 3.2. Different Metabolic Responses of FHB-Resistant and -Susceptible Wheats to Three Fusarium Strains

PCA of the NMR data for FHB-resistant and -susceptible wheat spikes showed that there were no obvious metabolic differences between those inoculated with mutants (including *TPS1*^−^ and *TPS2*^−^) and those inoculated with WT/ddH_2_O, and it was the same between those inoculated with *TPS1*^−^ and *TPS2*^−^ (Appendix A). Pairwise OPLS-DA was conducted between the extracts of wheat spikes inoculated with the *TPS*^−^ mutants and those inoculated with WT/ddH_2_O for both the resistant and susceptible wheat varieties. In addition, OPLS-DA modeling was also conducted comparing the extracts of wheat spikes inoculated with different *TPS^-^* mutants. Significantly different metabolites between these two groups were tabulated in Table 1. OPLS-DA model parameters showed that for the FHB-resistant wheat Sumai 3, there were clear metabolic differences between being inoculated with mutants (including *TPS1*^−^ and *TPS2*^−^) and with ddH_2_O/WT (Figure 3). Significantly altered metabolites between the two groups were tabulated in Table 1. The loadings plots of OPLS-DA showed that compared with ddH_2_O, inoculation with *TPS1*^-^ in the resistant wheat induced increased levels of Phe, uridine, guanosine, hypoxanthine, p-hydorxy cinnamic, acid and chlorogenic acid together with a decreased level of α-ketoglutarate (Figure 3a, Table 1). Inoculation with *TPS2*^−^ in Sumai 3 led to higher levels for Phe, guanosine, p-hydorxy cinnamic acid, and chlorgenic acid along with a lower level of α-ketoglutarate than with ddH_2_O, which was similar to inoculation with *TPS1*^−^; in addition, inoculation with *TPS2*^−^ also led to a decrease in phosphocholine level (Figure 3c, Table 1). Compared with WT, inoculation with *TPS1*^−^ in the FHB-resistant wheat resulted in elevated levels of guanosine and thymidine together with reduced levels of myo-inositiol, Ala, Asp, and glycine betaine (Figure 3b, Table 1). However, inoculation with *TPS2*^−^ in Sumai 3 led to increased thymidine level together with decreased levels of sucrose and myo-inositol (Figure 3d, Table 1). Furthermore, we compared the metabolic profiles for inoculation with the *TPS1*^−^ and *TPS2*^−^ mutants in the resistant wheat. The results indicated that there was no obvious metabolic difference between those inoculated with different *TPS*^−^ mutants (R^2^X = 0.636, Q^2^ = 0.126, CV-ANOVA *p* = 1).

For the susceptible wheat Annong 8455, OPLS-DA model parameters indicated that there were obvious metabolic differences between those inoculated with mutants (including *TPS1*^−^ and *TPS2*^−^) and with ddH_2_O (Figure 4). The color loading plots of OPLS-DA revealed that compared with ddH_2_O, infection with the *TPS1*^−^ mutant in the susceptible wheat induced increased levels of fumarate, glycine betaine, and adenosine (Figure 4a, Table 1). Infection with the *TPS2*^−^ mutant in Annong 8455 led to lower levels of sucrose, fumarate, α-ketoglutarate, phosphocholine, adenosine, guanosine, and thymidine (Figure 4b, Table 1). However, there were no significant metabolic differences between inoculation with *TPS1*^−^ and with WT (R^2^X = 0.868, Q^2^ = 0.258, CV-ANOVA *p* = 1) or between inoculation with *TPS2*^−^ and with WT (R^2^X = 0.769, Q^2^ = 0.0965, CV-ANOVA *p* = 1) in the susceptible wheat. The results also showed that for the susceptible wheat, compared with *TPS1*^−^, infection with *TPS2*^−^ resulted in a reduction in the levels of most metabolites including two sugars (glucose and fructose), Gln, three organic acids (fumarate, malate, and formate), two choline metabolites (glycine betaine and phosphocholine), and two nucleotide metabolites (adenosine and guanosine) (Figure 4c, Table 1).

## 4. Discussion

FHB induced by *Fg* is a destructive disease for wheat. In our previous study [21], we obtained two mutants, *TPS1*^−^ and *TPS2*^−^, carrying a single deletion of *TPS1* (trehalose 6-phosphate synthase) and *TPS2* (trehalose 6-phosphate phosphatase), respectively, both of these two enzymes being involved in trehalose synthesis in *Fg*. Liu et al. also found that *Fg TPS1*^−^ and *TPS2*^−^ both produce fewer mycotoxins than WT (5035), and *TPS2*^−^ produces fewer mycotoxins than *TPS1*^−^ [22]. However, the metabolic responses of FHB-resistant and -susceptible wheat to *Fg* (WT, *TPS1*^−^, and *TPS2*^−^) inoculation are not clear, but could give us metabolic information related to trehalose synthesis and FHB resistance.

Sumai 3 is a traditional FHB-resistant wheat variety, while Annong 8455 is a known FHB-susceptible wheat variety. The metabolic differences in both FHB-resistant and -susceptible wheat varieties when inoculated with *TPS*^−^ and with ddH_2_O could give us metabolic information about *Fg* pathogenicity, while the metabolic differences between inoculation with *TPS*^−^ and with WT in both FHB-resistant and -susceptible wheat varieties could give us metabolic information about the difference in pathogenicity induced by *TPS1* and *TPS2* mutations. 

For the resistant wheat Sumai 3, there were higher levels of Phe, guanosine, p-HCA and chlorogenic acid together with a lower level of α-KG after being inoculated with *TPS1*^−^ or *TPS2*^−^ when compared with being inoculated with ddH_2_O (Figure 3 and Figure 5, Table 1). This result indicated that Phe, p-HCA, and chlorogenic acid might be closely related to FHB-resistance. This is not surprising because both p-HCA and chlorogenic acid are secondary metabolites derived from the shikimate pathway for biosynthesis of phenylpropanoid and flavonoid metabolites. It has been reported that Sumai 3 has a fast and strong response primarily through the activation of the shikimate pathway [28]. HCAs have been reported to play roles in plant defense responses to pathogen challenge and wounding as integral components [46,47]. Chlorogenic acid, belonging to phenolic acids, has been reported to be a resistance factor to *Fg* in maize [48], to *Podosphaera* in miniature roses [49] and to diatraea saccharalis in sugarcane [50]. Sumai 3 might synthesize HCAs to reduce pathogen advancement through thickening host cell walls together with synthesizing antifungal/antioxidant chlorogenic acid for inhibiting pathogen growth, in turn reducing subsequent mycotoxin biosynthesis in *Fg* [51]. In a word, shikimate-mediated secondary metabolism was activated in the FHB-resistant wheat to produce HCAs and chlorogenic acid to inhibit the growth of *Fg* and reduce the production of mycotoxins.

In Sumai 3, when compared with that inoculated with WT, only the change trends of thymidine and myo-inositol were similar after being inoculated with *TPS1*^−^ or *TPS2*^−^ (Figure 3 and Figure 5, Table 1), which is attributed to the different impacts of *TPS1* and *TPS2* in *Fg* on wheat metabolism. The fact that the number of altered metabolites was less in *TPS2*^−^ vs. WT than in *TPS1*^−^ vs. WT suggested that the induced resistance response was much stronger in *TPS1*^−^ than in *TPS2*^−^, and the significantly decreased metabolites (Ala, Asp, and GB) might be closely related to FHB resistance in an unknown way (Figure 5). 

For FHB-susceptible wheat variety Annong 8455, there were no metabolic differences when inoculated with either *TPS1*^−^ or *TPS2*^−^ compared to inoculation with WT. This is not surprising because the metabolic responses of Annong 8455, a highly FHB-susceptible wheat variety, to *TPS1*^−^/*TPS2*^−^ and WT were similar, although the virulence of both *TPS1*^−^ and *TPS2*^−^ is lower than that of WT. For Annong 8455, compared to being inoculated with ddH_2_O, there were significantly higher levels of fumarate, GB and adenosine after being inoculated with *TPS1*^−^; however, there were significantly lower levels of sucrose, two organic acids (pyruvate and α-KG), phosphocholine, and three nucleic acids (adenosine, guanosine, and thymidine) after being inoculated with *TPS2*^−^ (Figure 4 and Figure 5, Table 1). This result indicated that *TPS1*^−^ infection induced a slight alteration in Annong 8455; nevertheless, *TPS2*^−^ infection slowed glycolysis, TCA cycle, and nucleotide synthesis. The former might be caused by trehalose deletion, and the latter might be caused by redundancy of T6P in *Fg* [21,52]. T6P in *Fg* may affect the global metabolism of wheat either as a metabolic regulator or through interaction with some compounds in wheat to regulate wheat metabolism [20]. However, how the redundancy of T6P in *Fg* affects wheat metabolism is still to be further studied.

## 5. Conclusions

In conclusion, when infected by *Fg*, FHB-susceptible and -resistant wheat showed different metabolic responses. Specifically, compared with ddH_2_O, resistant wheat increased the levels of Phe, p-HCA, and chlorogenic acid to resist *TPS*^−^ mutants; however, susceptible wheat did not. The metabolic difference might be caused by trehalose deletion and redundancy of T6P in the susceptible wheat when infected by *TPS1*^−^ or *TPS2*^−^ compared with ddH_2_O. Finally, a hypothesis is proposed that when infected by *Fg*, shikimate-mediated secondary metabolism was activated in the FHB-resistant wheat to produce HCAs and chlorogenic acid to inhibit the growth of *Fg* and reduce the production of mycotoxins. However, it should be stressed that the molecular relationships between the trehalose biosynthetic pathway in *Fg* and shikimate-mediated secondary metabolism in wheat remains to be further studied.

## Figures and Tables

**Figure 1 metabolites-12-00727-f001:**
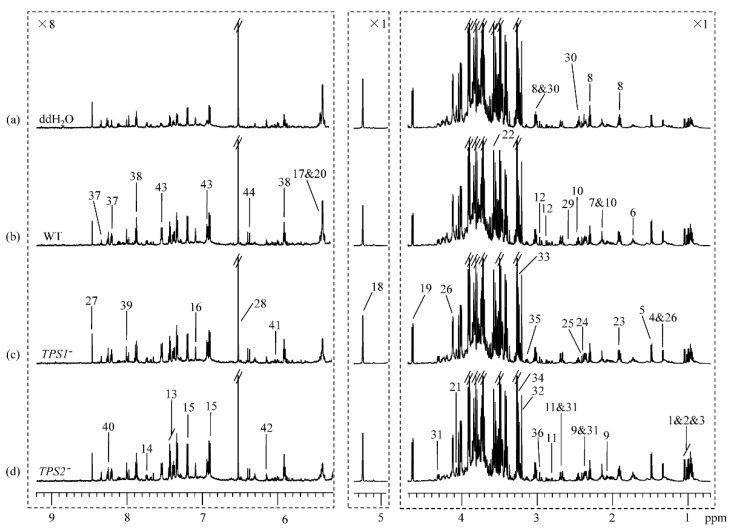
^1^H NMR spectra of extracts for resistant wheat Sumai 3 inoculated with (**a**) ddH_2_O, (**b**) *Fg* WT 5035, (**c**) *Fg TPS1*^−^ mutant, and (**d**) *Fg TPS2*^−^ mutant for 96 h. The region δ 5.31–9.21 was vertically expanded 8 times. Keys: 1, isoleucine (Ile); 2, leucine (Leu); 3, valine (Val); 4, threonine (Thr); 5, alanine (Ala); 6, arginine (Arg); 7, methionine (Met); 8, γ-aminobutyrate (GABA); 9, glutamate (Glu); 10, glutamine (Gln); 11, aspartate (Asp); 12, asparagine (Asn); 13, phenylalanine (Phe); 14, tryptophan (Trp); 15, tyrosine (Tyr); 16, histidine (His); 17, sucrose; 18, α-glucose; 19, β-glucose; 20, raffinose; 21, fructose; 22, myo-inositol; 23, acetate; 24, pyruvate; 25, succinate; 26, lactate; 27, formate; 28, fumarate; 29, citrate; 30, α-ketoglutarate (α-KG); 31, malate; 32, choline; 33, phosphocholine (PC); 34, glycine betaine (GB); 35, ethanolamine (EA); 36, dimethylamine; 37, adenosine; 38, uridine; 39, guanosine; 40, hypoxanthine; 41, inosine; 42, deoxy adenosine monophosphate (dAMP); 43, p-hydorxy cinnamic acid; 44, chlorogenic acid; 45, thymidine.

**Figure 2 metabolites-12-00727-f002:**
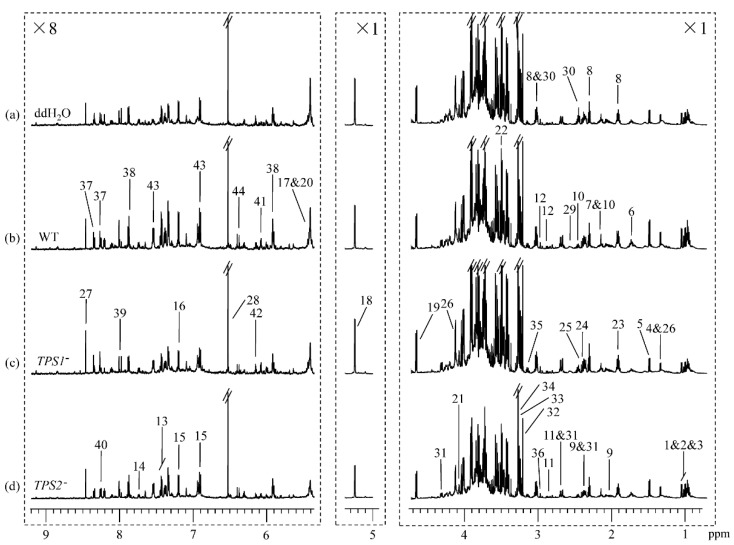
^1^H NMR spectra of extracts for susceptible wheat Annong 8455 inoculated with (**a**) ddH_2_O, (**b**) *Fg* WT 5035, (**c**) *Fg TPS1*^−^ mutant, and (**d**) *Fg TPS2*^−^ mutant for 96 h. The region *δ* 5.31–9.21 was vertically expanded 8 times. Keys were indicated in Figure 1 and Appendix A.

**Figure 3 metabolites-12-00727-f003:**
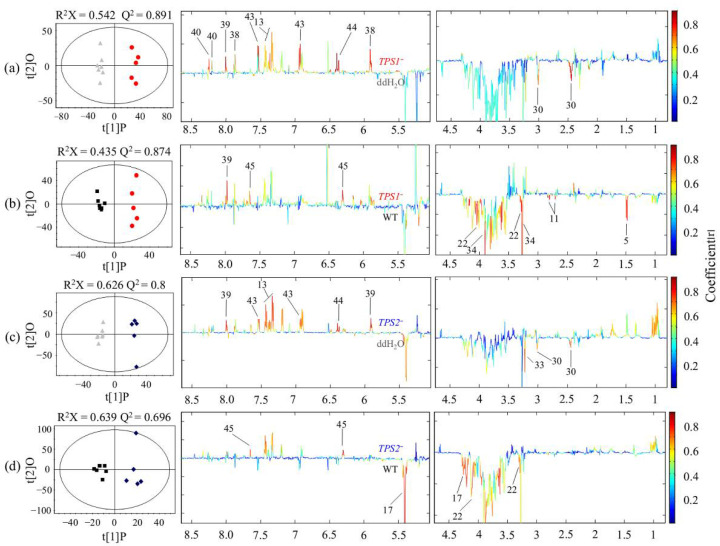
OPLS-DA scores plots (**left**) and coefficient-coded loadings plots (**right**) showing metabolic differences of FHB-resistant wheat Sumai 3 inoculated with (**a**) *TPS1*^−^ (red) vs. ddH_2_O (grey) (CV-ANOVA *p* = 0.0018), (**b**) *TPS1*^−^ (red) vs. WT (black) (CV-ANOVA *p* = 0.0029), (**c**) *TPS2*^−^ (blue) vs. ddH_2_O (grey) (CV-ANOVA *p* = 0.014), and (**d**) *TPS2*^−^ (blue) vs. WT (black) (CV-ANOVA *p* = 0.01). Metabolite keys are the same as in Figure 1 and Appendix A.

**Figure 4 metabolites-12-00727-f004:**
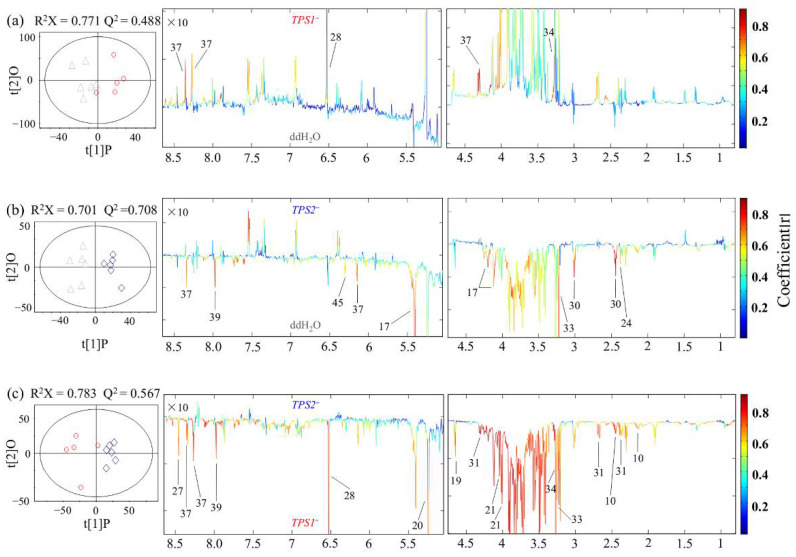
OPLS-DA scores plots (**left**) and coefficient-coded loadings plots (**right**) showing metabolic differences of FHB-susceptible wheat Annong 8455 inoculated with (**a**) *TPS1*^−^ (red) vs. ddH_2_O (grey) (CV-ANOVA *p* = 0.005), (**b**) *TPS2*^−^ (blue) vs. ddH_2_O (grey) (CV-ANOVA *p* = 0.001) and (**c**) *TPS2*^−^ (blue) vs. *TPS1*^−^ (red) (CV-ANOVA *p* = 0.005). Metabolite keys are the same as in Figure 1 and Appendix A.

**Figure 5 metabolites-12-00727-f005:**
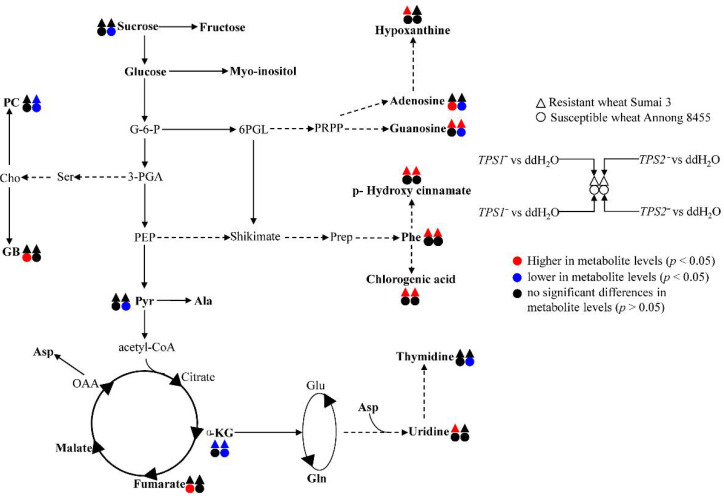
The metabolic responses of Sumai 3 and Annong 8455 to *TPS*^−^ mutants and ddH_2_O inoculation. Red colored symbols indicate significant up-regulations of metabolites (*p* < 0.05), whereas blue colored symbols represent downregulations of metabolites (*p* < 0.05). G-6-P, glucose-6-phosphate; 3-PGA, 3-phosphate glycerate; Ser, serine; Cho, choline; PC, phosphocholine; GB, glycine betaine; OAA, oxaloacetic acid; Pyr, pyruvate; PEP, phosphoenolpyruvic acid; 6PGL, 6-phosphogluconate; PRPP, 5-phosphoribosyl diphosphate; Prep, prephenic acid.

**Table 1 metabolites-12-00727-t001:** Significantly changed metabolites in the resistant and susceptible wheat when inoculated with ddH_2_O (H_2_O), WT, *TPS1*^−^, and *TPS2*^−^.

	Coefficient (r) *^a^*						
Metabolites (No)	Sumai 3 (Resistant)				Annong 8455 (Susceptible)		
**Sugars**	*TPS1*^−^ vs. H_2_O	*TPS1*^−^ vs. WT	*TPS2*^−^ vs. H_2_O	*TPS2*^−^ vs. WT	*TPS1*^−^ vs. H_2_O	*TPS2*^−^ vs. H_2_O	*TPS2*^−^ vs. *TPS1*^−^
sucrose (17) *^c^*				−0.849 *^b^*		−0.882	
α-glucose (18)							−0.826
β-glucose (19)							−0.816
fructose (21)							−0.857
myo-inositol (22)		−0.788		−0.821			
**Amino acids**							
Ala (5)		−0.781					
Gln (10)							−0.762
Asp (11)		−0.857					
Phe (13)	0.835		0.756				
**Organic acids**							
pyruvate (24)						−0.841	
formate (27)							−0.823
fumarate (28)					0.794		−0.904
a-KG (30)	−0.836		−0.782			−0.769	
malate (31)							−0.963
**Choline metabolites**							
phosphocholine (33)			−0.859			−0.877	−0.871
glycine betaine (34)		−0.837			0.814		−0.892
**Nucleotide metabolites**							
adenosine (37)					0.824	−0.805	−0.917
uridine (38)	0.825						
guanosine (39)	0.929	0.758	0.799			−0.826	−0.799
hypoxanthine (40)	0.856						
thymidine (45)		0.821		0.834		−0.843	
**Secondary metabolites**							
p-hydroxy cinnamaic acid (43)	0.854		0.753				
chlorogenic acid (44)	0.922		0.826				

*^a^* The coefficients were obtained from OPLS-DA results, and positive and negative signs indicate positive and negative correlation in the concentrations, respectively. *^b^* Positive and negative signs indicate the elevation and decrease of the metabolite levels. Values for *p* ≥ 0.05 were not tabulated. *^c^* Metabolite keys are identical to those in Figure 1 and Appendix A.

## Data Availability

Not applicable.

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
