# Peer review of "The Different Metabolic Responses of Resistant and Susceptible Wheats to Fusarium graminearum Inoculation"

_metabolites, 2022, doi:10.3390/metabo12080727_

Round 1

Reviewer 1 Report

Manuscript title: The different metabolic responses of resistant and susceptible wheats to Fusarium graminearum inoculation

Manuscript ID:  metabolites-1826451

Journal:  Metabolites

The main objective of the current manuscript was to analyze the metabolomics profile of FHB-resistant and -susceptible wheat varieties inoculated with ddH2O, WT, TPS- mutants. The paper is well organized (especially material and methods section), written and gave me direct idea about the subject. In the abstract please insert the most important results in values and/or percentage. Material and methods contain details which help other researchers to follow. However, high resolution figures must be provided for results. Discussion section needs to me improved focusing on your findings.  

Author Response

We are grateful for the viewer’s work and have responded to the comments as follows in a point-by-point fashion with some corrections marked in the manuscript (in red):  

Q1: In the abstract please insert the most important results in values and/or percentage.

Reply: Accepted and attended. We have corrected into “Shikimate-mediated secondary metabolism was activated in the FHB-resistant wheat to inhibit the growth of Fg and reduce the production of mycotoxins” in the manuscript.

Q2: However, high resolution figures must be provided for results

Reply: Accepted and attended. High resolution figures (300 dpi) have been provided in the latest manuscript.

Q3: Discussion section needs to me improved focusing on your findings.  

Reply: Accepted and attended. In the discussion section, we add one sentence “In a word, shikimate-mediated secondary metabolism was activated in the FHB-resistant wheat to produced HCAs and chlorogenic acid to inhibit the growth of Fg and reduce the production of mycotoxins.” in the discussion section. Meanwhile, we add corresponding sentences in the conclusions.

Apart from the above, we have also made some minor corrections in the manuscript including references.

Reviewer 2 Report

The authors are experienced Fusarium graminearum metabolism researchers. In here they study 2 varieties with different FHB resistances and how they are affected by  TPS1/TPS2 gene deletions.  The analytical approaches are well explained and the NMR metabolite identification procedure is described in detail. 

The major issue is the PLS-DA design, specifically the CV process. The number of samples seem to be 5-7 per experimental condition, such number of samples should not be used for a K-fold CV, and thus a LOO validation might be more suitable. This affects model quality. 

Next, in the first lines of the Results sections, the authors state a "obvious difference" between Figure 1 and Figure 2. These could be reported better in a colored figure of somekind as, the differences are only noticeable in certain regions of the spectra.

Other comments:
- There is probably a word missing in line 53 
- "obtained" probably typo in line 279

Author Response

We are grateful for the review’s work and have responded to their comments as follows in a point-by-point fashion with some corrections marked in text (in red):

Q1: The major issue is the PLS-DA design, specifically the CV process. The number of samples seem to be 5-7 per experimental condition, such number of samples should not be used for a K-fold CV, and thus a LOO validation might be more suitable. This affects model quality. 
Reply: Accepted and attended. LOO validation was further used in the OPLS-DA models. The values of Q2 and p of CV-ANOVA showed that the quality of models was better (Loo validation). However, the loading plots of models were same to previous models (K-fold). So we have updated the corresponding values in the figures and manuscript.

Q2: Next, in the first lines of the Results sections, the authors state a "obvious difference" between Figure 1 and Figure 2. These could be reported better in a colored figure of somekind as, the differences are only noticeable in certain regions of the spectra

Reply: Good point. We are so sorry for that we do not describe clearly, and we do not make a comparison between Figure 1 and Figure 2 in the first section of results. So we have corrected into “1H NMR spectra of wheat spike extracts showed obvious difference in metabolic profiles for FHB-resistant wheat (Sumai 3) inoculated with ddH2O, WT, TPS1- and TPS2- (Figure 1). Similarly, there was also obvious difference in metabolic profiles for the FHB-susceptible wheat (Annong 8455) inoculated with ddH2O, WT, TPS1- and TPS2- (Figure 2) ’’ in the manuscript. In addition, the edition of TopSpin we used might be a little old, and we could not obtain color figures from “plot” order. 

Q3: Other comments:
- There is probably a word missing in line 53   
- "obtained" probably typo in line 279   

Reply: Accepted and attended.

Line 53 ---We have corrected “metabolomics” into “metabonomics”. In addition, we also made some corrections in the other part of the text about this word.

Line 279 ---We can’t find "obtained" in line 279. We also search "obtained" in the other places of the text, and check carefully.

Apart from the above, we have made a thorough overhaul to improve the manuscript grammatically and logically, and we have also checked carefully the format of the references.
